# Influence of the Volatility of Solvent on the Reproducibility of Droplet Formation in Pharmaceutical Inkjet Printing

**DOI:** 10.3390/pharmaceutics15020367

**Published:** 2023-01-21

**Authors:** Robert Mau, Hermann Seitz

**Affiliations:** 1Microfluidics, Faculty of Mechanical Engineering and Marine Technology, University of Rostock, Justus-von-Liebig Weg 6, 18059 Rostock, Germany; 2Department Life, Light & Matter, Interdisciplinary Faculty, University of Rostock, Albert-Einstein-Str. 25, 18059 Rostock, Germany

**Keywords:** pharmaceutical piezoelectric inkjet printing, drop-on-demand (DOD), droplet formation, drug solutions, solvent volatility, solvent evaporation, creeping, crystallization

## Abstract

Drop-on-demand (DOD) inkjet printing enables exact dispensing and positioning of single droplets in the picoliter range. In this study, we investigate the long-term reproducibility of droplet formation of piezoelectric inkjet printed drug solutions using solvents with different volatilities. We found inkjet printability of EtOH/ASA drug solutions is limited, as there is a rapid forming of drug deposits on the nozzle of the printhead because of fast solvent evaporation. Droplet formation of c = 100 g/L EtOH/ASA solution was affected after only a few seconds by little drug deposits, whereas for c = 10 g/L EtOH/ASA solution, a negative affection was observed only after t = 15 min, while prominent drug deposits form at the printhead tip. Due to the creeping effect, the crystallizing structures of ASA spread around the nozzle but do not clog it necessarily. When there is a negative affection, the droplet trajectory is affected the most, while the droplet volume and droplet velocity are influenced less. In contrast, no formation of drug deposits could be observed for highly concentrated, low volatile DMSO-based drug solution of c = 100 g/L even after a dispensing time of t = 30 min. Therefore, low volatile solvents are preferable to highly volatile solvents to ensure a reproducible droplet formation in long-term inkjet printing of highly concentrated drug solutions. Highly volatile solvents require relatively low drug concentrations and frequent printhead cleaning. The findings of this study are especially relevant when high droplet positioning precision is desired, e.g., drug loading of microreservoirs or drug-coating of microneedle devices.

## 1. Introduction

Drop-on-demand (DOD) inkjet printing is an attractive material deposition and patterning technology with a high degree of automation and scalability. As a result, there is a wide range of applications in the industrial and scientific environment. Originally utilized for digital printing of (colored) inks for print media, DOD has emerged as a well-established tool in multiple fields. It is utilized in the fields of additive manufacturing (3D printing) [1], bioprinting [2,3], manufacturing of flexible (bio)sensors [4,5], high-throughput screening and micro arraying for biotechnological/pharmaceutical research [3] or even for display manufacturing [6], as some representative examples.

Furthermore, over the recent decade, inkjet printing technology has offered promising paths for the pharmaceutical industry, especially for manufacturing new pharmaceutical dosage forms to deliver drugs [7,8,9,10,11,12,13]. A milestone in the field of alternatives for conventional dosage forms was the FDA-approval of Spritam^®^ in 2015 [14,15], as the first marketed 3D printed tablet of levetiracetam, which is manufactured via an inkjet-based 3D printing process. Furthermore, inkjet printing is a promising tool for manufacturing various kinds of other drug delivery systems. For example, it has been used for drug loading of microneedles for transdermal drug release [16,17,18], coating and drug loading of drug-eluting stents [19], drug loading of reservoir-based systems for oral drug delivery [20,21,22] or drug laden particles [23]. Moreover, inkjet printing enables innovative hybrid 3D printing methods for more precision in drug loading of 3D printed drug delivery systems [24] or to incorporate local drug depots [25]. In sum, inkjet printing offers much potential to push the innovation portfolio for further patient specific treatments with reduced toxicity and improved efficacy, safety and patient compliance [26,27].

For inkjet printing of an active pharmaceutical ingredient (API), there is the need to formulate an inkjet-suitable drug solution. The desired API has to be dissolved or dispersed, respectively, in a carrier liquid, a solvent, to enable droplet formation. The result of an inkjet printing process depends on the droplet formation, the droplet impact on the substrate as well as the drying and solidification of the droplets after printing. To estimate, respectively, the droplet formation and the printability of a liquid in an inkjet process, the *Z* number (Formula (1)) can be used. It is the inverse of the well-known Ohnesorge number (*Oh*) and in recent years *Z* is increasingly replacing *Oh* [28]. The *Z* number relates to the inertial and surface tension forces to viscous forces. In our study, we use the diameter of the printhead nozzle *d_N_* as a characteristic length. This is common practice [29] and, moreover, it enables pre-evaluation of inkjet printability of a fluid before performing inkjet printing experiments. If *Z* is too low, viscous forces prevent a droplet’s separation from the nozzle of the printhead. If *Z* is too high, then droplet formation may show high numbers of satellite droplets, which may be disadvantageous for the printing resolution. There are different recommendations for an appropriate interval of *Z* for inkjet applications, such as 1 < *Z* < 10 [30], 4 < *Z* < 14 [31] or 2 < *Z* < 20 [28]. This underlines that there is a need for the experimental validation of each combination of an inkjet printing setup and the liquid which is printed. The Weber number (*We*, Formula (2)) can be used to estimate the droplet impact on the substrate. It relates the inertia forces to surface tension. To investigate the droplet impact on a surface it is common practice to use the droplet diameter *d_D_* as a characteristic length [32]. If *We* is too high (*We* ≥ 100 on solid surfaces, *We* ≥ 40 on wetted surfaces), then the droplet impact is characterized as strong [32]. In this case, the jetted droplets tend to splash when impacting on the surface. Consequently, secondary droplets would eject from the impacting main droplet, which is negative for precise droplet positioning. For high droplet positioning precision, it is desirable to aim for smaller values of *We*, since droplet impact is defined as weak (*We* < 100 on solid surfaces, *We* < 40 on wetted surfaces) [32].
(1)$$Z=\frac{1}{Oh}=\frac{\sqrt{\gamma \rho d_{N}}}{\eta }$$
(2)$$We=\frac{u_{D}{}^{2}\rho d_{D}}{\gamma }$$

*Z*—*Z* number*Oh*—Ohnesorge number*We*—Weber number*γ*—surface tension*ρ*—density*η*—dynamic viscosity*u_D_*—velocity of droplet at flight*d_N_*—diameter of nozzle*d_D_*—diameter of droplet

Since the results of inkjet printing depend on the drying of the droplets and solidification of the API after printing, the role of the solvent is not only to act as a carrier liquid for a suitable droplet generation but also to evaporate at a controlled rate after printing to deliver the API in an appropriate solid form [29]. The evaporation process depends significantly on the volatility of a solvent. The volatility of a solvent is described in the German industrial standard DIN 53170 via a dimensionless number—the evaporation number. That number is determined by the ratio of evaporation time of a specific volume of a specific fluid to the evaporation time of the same specific volume of diethyl ether. The evaporation number of a highly volatile solvent is low (<10), whereas the evaporation number of a low volatile solvent is high (>50) [33,34]. The volatility of a solvent of a drug solution plays a significant role, especially when dealing with nucleation and crystal growth of an API during the drying process of printed droplets [7,9]. Moreover, the volatility of the solvent of a drug solution has an impact on droplet formation from the printhead. High volatilities may be beneficial for fast inkjet printing, as the solvent of the printed droplets of the drug solution quickly evaporates from the substrate, leaving behind the drug to be delivered. Especially, applications such as the filling of reservoirs can benefit from high evaporation rates [35]. 

However, there is the risk of nozzle clogging because of drug deposits from evaporating drug solution on the nozzle of the printhead, especially when using highly volatile solvents [7,35,36]. In this context, the drug concentration in the drug solution also plays a crucial role. Higher drug concentrations can be beneficial for minimizing the fabrication time. On the other hand, the stability and printability of the solution may be compromised [29]. Moreover, crystallizing substances, which include most API, can show creeping when crystallizing from evaporating solutions. Townsend et al. define creeping as the evaporation-driven extension of crystals from the boundary of a solution on a solid nonporous substrate [37]. Widespread crystalline structures can appear after droplet evaporation of a solution of a crystallizing substance. These structures can spread out over an area larger than the wetted area of the initial droplet on the substrate. Creeping is differentiated in primary and secondary creeping. Primary creeping is defined as the direct growth of the crystals on the substrate surface [38], leading to relatively flat, widespread crystalline structures on a surface. Since the extension of the crystals starts from the boundary of the liquid phase on the surface of the substrate, the crystalline structures typically grow and spread around the initially wetted area, directed away from the center of the liquid phase [37,39]. Secondary creeping is defined as creeping upon previously deposited crystals [38]. It follows after primary creeping and leads to distinct crust-like crystalline structures, that can spread strongly in all spatial directions as shown in [38].

The risk of nozzle clogging due to drug deposits from the evaporating drug solution is known from the literature [7]. However, the current literature lacks knowledge about the influence of drug deposits on the nozzle of a printhead on the various parameters of droplet formation such as droplet volume, droplet speed, and the trajectory of the droplets during the time that droplet formation still occurs before the nozzle becomes clogged. Moreover, it is unclear how creeping affects nozzle clogging. Since the creeping-driven extension of crystals start from the boundary of a solution, distinct drug deposits can form around the nozzle of a printhead, which need not necessarily clog it. The volatility of the solvent and the drug concentration in the solution are important factors in the crystallization of drugs from an evaporating drug solution [29]. Consequently, in this study, we investigate inkjet printing of different concentrated, highly volatile drug solutions based on ethanol (EtOH) and Acetylsalicylic acid (ASA). Moreover, we compare the inkjet printing of these EtOH drug solutions to the inkjet printing of dimethyl sulfoxide (DMSO)-based drug solutions, as DMSO has a significantly lower evaporation rate as EtOH. EtOH is highly volatile (evaporation number of 8.3 [34]), whereas DMSO has a very low volatility (evaporation number of ≈700 [34]). Both EtOH and DMSO are frequently used in pharmaceutical inkjet printing [29], but there is a lack of knowledge about the effects of their different evaporation behaviors on droplet formation in the inkjet printing process. In our study, printhead operating parameters for a successful droplet formation are investigated for a piezoelectric drop-on-demand (DOD) inkjet printhead in a first step. In a second step, long-term droplet formation investigations are performed with preferable printhead driving parameters. In our study, we use the *Z* number to evaluate the inkjet printability of the drug solutions based on fluid properties. In addition, we use the *We* number to define preferred printhead operating parameters based on the droplet impact on the surface. Regarding the choice of the model drug, in the literature, there is no focus on any specific API for pharmaceutical inkjet printing. In the past, small molecule drugs were frequently used [29]. Most API are small molecule drugs, as they have an average molecular weight below 550 Da [40]. ASA fits with that with its average molecular weight of 180.16 Da [41]. Furthermore, ASA was used in previous studies as a model drug to investigate crystallization behavior because its chemical structure corresponds to the chemical structure of many other APIs [42,43]. Since drug deposits at the printheads nozzle are expected due to drug crystallization by solvent evaporation, ASA is a suitable model drug for our study. Moreover, ASA shows high solubility in the solvents EtOH and DMSO [44].

Our study helps to decide on a solvent (with regard to the volatility) and the level of drug concentration in solution for (long-term) inkjet printing. Moreover, our study provides detailed findings about the growth of drug deposits on the tip of the printhead over time and its influence on significant droplet parameters.

## 2. Materials and Methods

### 2.1. Preparation of Drug Solutions

Three different drug solutions were prepared as listed in Table 1 based on the solvents EtOH (≥99.8%; Carl Roth GmbH + Co. KG, Karlsruhe, Germany) and DMSO (≥99.7%; Merck KGaA, Darmstadt, Germany) as well as ASA as a model drug (≥99.0%; CAS No.: 50-78-2; Merck KGaA, Darmstadt, Germany). ASA was solved in the solvents at an ambient atmosphere using a magnetic stirrer until there was a clear drug solution. Two different concentrated drug solutions based on EtOH were prepared. Because of the high volatility of EtOH, we expect the droplet formation to be affected by crystallizing drug deposits on the nozzle during the printing process. Consequently, in addition to a relatively high concentrated drug solution (EtOH100ASA: c = 100 g/L), we prepared a relatively low concentration drug solution (EtOH10ASA: c = 10 g/L).

### 2.2. Characterization of Drug Solutions and Analysis of the Inkjet Printability

The density *ρ* of the drug solutions is measured using a density meter DSA 5000 M (Anton Paar GmbH, Graz, Austria) at a temperature of T = 21 ± 0.05 °C (temperature controlled via DSA 5000 M device). The surface tension *σ* is measured using the contact angle and droplet contour analysis system OCA 40 Micro (DataPhysics Instruments GmbH, Filderstadt, Germany) at ambient atmosphere (T = 21 ± 2 °C). The dynamic viscosity *η* is measured using a Haake Mars II rheometer equipped with the double cone measurement system DC 60/1° (Thermo Fisher Scientific Inc., Waltham, MA, USA) at a shear rate of γ = 250 s^−1^ and at a temperature of T = 21 °C ± 1 °C (temperature controlled via Haake Mars II rheometer). All measurements are repeated three times. The inkjet printability of the drug solutions is analyzed using the *Z* number. The nozzle diameter *d_N_* = 29 µm of the PicoTip printhead is used to calculate *Z* number via Formula (1). The nozzle diameter is determined with a microscope of type BX51 (Olympus K.K., Tokyo, Japan) using the software “analySIS docu” (version 5.1, Olympus K.K., Tokyo, Japan).

### 2.3. Inkjet Printing Device with Droplet Formation Analyzation Setup

Experimental inkjet printing investigations are performed via the nano/picoliter inkjet system Nanoplotter 2.1 (hereafter referred to as nanoplotter, Figure 1) equipped with a piezo-driven drop-on-demand printhead “Pico-Tip J” (both GeSiM, Gesellschaft für Silizium-Mikrosysteme mbH, Radeberg, Germany). The Pico-Tip J printhead enables dispensing of droplets of a typical size of about 50 pl up to 80 pl. The used printhead has a nozzle diameter of *d_N_* = 29 µm. The nanoplotter is equipped with a stroboscopic image capturing system for droplet parameter analysis. Moreover, the nanoplotter features a washing station. It enables cleaning of the printheads from the inside and the outside with water.

### 2.4. Investigation of Inkjet Printing Parameters

For each drug solution we investigate preferable inkjet printing parameters to enable long-term droplet formation investigations. The droplet parameters droplet volume *V_D_*, droplet speed *u_D_* and the direction of the droplet trajectory, described as deviation angle α from a reference line, are investigated as a function of voltage U of the printhead. Since we use a piezoelectric inkjet printhead, the voltage of the printhead is the most significant parameter to affect the droplet parameters. We vary the voltage, which is used to operate the printhead, from U = 35 V to U = 80 V via steps of 5 V. Fix values were set for the operation parameters of pulse length t_pulse_ = 20 ms and frequency f = 100 Hz as these are recommended by the manufacturer to be used for the Pico-Tip J printhead. A number of *n* = 10 measurements is performed for each droplet parameter over a time of t = 15 s via the stroboscopic image capturing system of the nanoplotter.

The nanoplotter enables the differentiation of the deviation angle of the main droplets *α_MD_* and the deviation angle of the satellite droplets *α_SD_*. The parameter *α_MD_* is defined as the deviation angle of the main droplet’s trajectory from a nozzle vertical line. The parameter *α_SD_* is defined as the deviation angle of the trajectory of the satellite droplets from the trajectory of the associated main droplet. For the parameter of *V_D_*, there is no differentiation of main and satellite droplets, as the algorithm of the stroboscopic image capturing system measures the total volume of liquid ejected during a single droplet generation process. The image capturing system does not allow the volume of the main droplet and satellite droplets to be measured separately. The parameter of droplet speed *u_D_* describes the speed of the main droplet.

To select a suitable printhead voltage for long-term droplet formation investigations, the *We* number (Formula (2)) will be used. The *We* number is calculated from the determined fluid parameters and the droplet diameter *d_D_* of the main droplets as the characteristic length. That conforms with the assumption that the droplets are of ideal spherical geometry, which is typical for the small droplet sizes in inkjet printing [45]. The parameter *d_D_* is calculated from the average of the measured values of droplet volume V_D_. The investigations are performed in ambient atmosphere (air, T = 21 °C ± 2 °C, relative humidity of 50% ± 10%).

### 2.5. Investigations of Long-Term Droplet Formation

After defining inkjet printing parameters for each drug solution, long-term droplet formation is investigated. Since DOD inkjet printing ejects fluid from the printhead only when a drop is required, it is frequently used for high precision dosing/printing applications when a relatively small number of individual droplets are required over a relatively long period of time. Therefore, the period of time is varied to investigate the time-dependent droplet formation. In a dispensing test series, 500 droplets each are ejected uniformly over each of four different time periods (t):
t = 1 min,t = 4 min,t = 15 min andt = 30 min.

A voltage of the printhead of U = 47 V for EtOH-based drug solution and U = 57 V for DMSO-based drug solution as well as a t_pulse_ of 20 ms were chosen. These operating parameters were chosen on the basis of the results of the investigations from Section 2.4 and ensure, on the one hand, a suitable energy input for droplet formation and, on the other hand, a uniform droplet volume of *V_D_* ≈ 50 pl for the different drug solutions in order to achieve comparability of the results. The droplet formulation directly before and directly after each dispensing test is analyzed via the stroboscopic image capturing system of the nanoplotter (*n* = 5). The parameters of droplet volume *V_D_* and the deviation angles *α_MD_* and *α_SD_* are investigated. Between every dispensing test, the tip of the printhead is cleaned with distilled water from inside and outside (depth of immersion in water ≈ 5 mm) for a time of t_wash_ = 30 s with an automated wash procedure in the washing station of the nanoplotter. The investigations are performed in ambient atmosphere (air, T = 21 °C ± 2 °C, relative humidity of 50% ± 10%). For statistical reasons, each dispensing test is repeated ten times.

### 2.6. Statistical Analysis

The details of all statistical analyses are given in the table and figure captions. The plotting of the graphs and statistical analyses are performed using the software OriginPro 2018b (OriginLab Corporation, Northampton, MA, USA). If not described otherwise, means (arithmetic average) ± standard deviations are used for data analysis. The fluid properties of the solvents and drug solutions (density *ρ*, surface tension *σ* and dynamic viscosity *η)* are measured *n* = 3 each. The nozzle diameter of the printhead is measured once. The *Z* number is calculated from the result of the single measurement of the nozzle diameter and the mean values of the fluid properties. For inkjet printing parameter investigations, each droplet parameter was measured ten times. For long-term droplet formation investigations, each droplet parameter was measured fifty times.

## 3. Results and Discussion

### 3.1. Properties of Drug Solutions and Inkjet Printability

Table 2 shows the values for the measured density *ρ*, surface tension *σ* and dynamic viscosity *η* as well as the calculated values for *Z* of the drug solutions and of the pure solvents. The values of these fluid parameters are higher for the ASA drug solutions than for the pure solvents. Nevertheless, the fluid parameters of the tested drug solutions remain suitable for inkjet printing. The values of *Z* range from 2 to 20, which has been shown to be suitable for inkjet printing [28]. However, the values of *Z* are relatively high, so the appearance of satellite droplets is likely [31,46].

### 3.2. Inkjet Printing Parameters

Figure 2 shows exemplary photos of droplet formation of the investigated drug solutions EtOH100ASA, EtOH10ASA and DMSO100ASA over a time of t = 15 s, recorded with a stroboscopic image capturing system of the Nanoplotter. All the investigated drug solutions show a droplet formation with one satellite droplet in these examples. The droplet volume is about *V_D_* = 45 pl for EtOH100ASA and ETOH10ASA), and about *V_D_* = 55 pl for DMSO100ASA. The printhead was operated with a voltage of U = 45 V for EtOH100ASA and EtOH10ASA both, and U = 60 V for DMSO100ASA. The reproducibility of droplet formation of EtOH100ASA is negatively affected by drug deposits on the nozzle of the printhead. These increase steadily over the time course of the investigation (Figure 2A1–A4). Especially, the reproducibility of the droplet’s trajectory is affected negatively, because the drug deposits change the nozzle’s geometry and wetting behavior, which are significant factors for droplet formation, respectively, the droplet pinch-off from the nozzle [47]. In contrast, we found no drug deposits for EtOH10ASA and DMSO100ASA. As a result, there is a stable, homogeneous droplet formation. At the beginning of the investigation it looks almost the same as at the end of the investigation after a time of t = 15 s (Figure 2B1,B2,C1,C2).

Figure 3 shows the plot of the droplet’s trajectory, described via the averaged values ± standard deviation of deviation angle of main droplets *α_MD_* and satellite droplets *α_SD_* and plotted over voltage of the printhead. EtOH10ASA and pure EtOH show similar small averages of *α_MD_* and *α_SD_* nearly around 0° and small standard deviations of <±1°, when there is an adequate energy input of U > 45 V. Even with an adequate energy input of U > 55 V, pure DMSO and DMSO100ASA show more fluctuation of the averages between values of ≈2° and ≈−2°, but the standard deviations are also small (<±1°), and this indicates a stable droplet formation. For EtOH100ASA, the results are different because of the formation of drug deposits on the nozzle of the printhead (exemplarily shown in Figure 2). There are relatively high standard deviations with fluctuating maxima over the whole voltage range. There is, thus, low reproducibility of a droplet’s trajectory even over the relatively short period of investigation of t = 15 s, because the droplet pinch-off is affected as wetting conditions at the tip of the printhead are not constant due to the growth of drug deposits.

Figure 4A plots the ejected droplet volume *V_D_* and Figure 4B plots the droplet speed *u_D_* as a function of the voltage (means ± standard deviations). In comparison to EtOH-based drug solutions, the DMSO-based solutions need a higher voltage U for a similar droplet volume and droplet speed. As shown before, in Table 2, EtOH has a lower viscosity and a lower surface tension than DMSO. As a result, there is a need for more energy input to induce droplet formation for DMSO. Whereas a minimum of U = 35 V voltage is needed to dispense pure EtOH or the EtOH-based drug solutions, a minimum of U = 45 V is necessary for pure DMSO and the DMSO-based drug solution.

Despite there being drug deposits forming on the nozzle of the printhead, the reproducibility of both parameters is not affected significantly for EtOH100ASA. The standard deviations of *V_D_* and *u_D_* are relatively low, which is in contrast to the standard deviations of *α_MD_* and α_SD_. The reproducibility of *V_D_* and *u_D_* is high. This indicates that there is no significant nozzle clogging over the time of investigation, despite the growth of considerable drug deposits. The reason for this is that the expansion of the crystals starts from the boundary of the liquid phase (at the edge of the nozzle) and is mostly directed away from the center of the liquid phase, respectively, the center of the nozzle. This is typical of the effect of creeping [37,39].

The mean values of *V_D_* and *u_D_* of EtOH100ASA are slightly below those of pure EtOH and EtOH10ASA. There is a significant increase of viscosity when solving 100 g/L of ASA in EtOH (see Table 2). So the decrease of *V_D_* and *u_D_* is quite plausible and in accordance with the literature [28]. Viscosity is related to the internal friction of a liquid, so there is a need for higher energy input for equal volumes, to be ejected from the nozzle.

Figure 5 shows a line graph of *We* over the voltage U of the printhead. DMSO-based fluids show lower *We* values than EtOH-based fluids at the same voltages because of the differences of the fluid parameters of significantly higher viscosity and surface tension.

When comparing the pure solvents and the drug solutions, we found relatively little differences between EtOH and EtOH10ASA, respectively, DMSO and DMSO100ASA, as the influence of drug solvation on fluid and drop formation parameters is small. However, in the case of EtOH100ASA, the *We* is significantly lower, which is mostly a consequence of a decrease of the droplet speed *u_D_*, as it is squared in Formula (2).

Nevertheless, a decrease of *We* is not a critical factor for droplet formation as low *We* are preferable over high *We* for a desirable droplet impact without splashing effects. *We* < 100 (droplet impact on solid substrates), respectively *We* < 40 (droplet impact on wetted surfaces) are useful to avoid the splashing of droplets on the substrate [32], which is beneficial for droplet positioning precision. Nevertheless, the lack of reproducibility of *α_MD_* and α_SD_, as we found for EtOH100ASA, is negative for droplet positioning precision.

As droplet formation of EtOH10ASA and DMSO100ASA was highly reproducible, we chose these two drug solutions for further long-term droplet formation investigations. EtOH100ASA is discarded because of the lack of reproducibility of the droplet’s trajectory. Moreover, we chose a voltage for the printhead of U = 47 V for EtOH10ASA and U = 57 V for DMSO100ASA for further investigations. Because of higher viscosity and surface tension, there is a need for more energy input for DMSO100ASA than for EtOH10ASA to ensure a uniform droplet volume of *V_D_* ≈ 50 pl. Moreover, these voltages are high enough to ensure a successful, homogenous droplet formation. On the other hand, these voltages are low enough to ensure a relatively low *We* < 40, which is beneficial for the impact of a droplet on a substrate.

The occurrence of satellite droplets is likely. In general, the formation of satellite droplets should be avoided because they can affect the precision of inkjet printing and thus hinder its applicability. However, the presence of satellite droplets can be tolerated if the satellite joins the main droplet on the surface [29]. For this purpose, using the preferred printhead parameters, the angular deviation of the satellite droplet trajectory from the main droplet trajectory should be minimal, as described via *α_SD_* (see Figure 3B). However, the literature also describes ways that can help to avoid the formation of satellite droplets. These include, for example, changing the rheology of the liquid or adjusting the electrical signal used to drive the printhead [29,48].

### 3.3. Long-Term Droplet Formation

Figure 6 shows exemplary images of droplet formation for EtOH10ASA (A) directly before and directly after the dispensing tests with a duration of (B) t = 1 min, (C) t = 4 min, (D) t = 15 min and (E) t = 30 min. A growing formation of drug deposits all over the tip of the printhead with an increasing time period can be observed. For the time periods t = 1 min, t = 4 min and t = 15 min, none of the ten experimental replicates resulted in the discontinuation of droplet formation, but droplet formation is affected in the dispensing test with the time period of t = 15 min. In the dispensing test with the time period of t = 30 min, clogging of the nozzle and termination of droplet formation occurred in 8 out of 10 experimental replicates. Droplet formation still occurred in 2 out of 10 experimental replicates but was significantly impaired (exemplarily shown in Figure 6E2). Due to the reduced droplet speed, there are no more satellite droplets. The parameter of *u_D_* was not measured during the long-term droplet formation investigations, but it is noticeable that the distance of the droplet from the tip of the printhead is shorter at the end than at the beginning of the dispensing tests (exemplarily shown in Figure 6A). Moreover, the flight direction of the droplet has changed significantly. The automated washing procedure of the nanoplotter with water (t_wash_ = 30 s) after a dispensing test was able to clean the tip of the printhead completely from the drug deposits. In the case of DMSO100ASA, even in the dispensing test with the time period of t = 30 min, no drug deposits were detected on the printhead tip (Figure 7). The droplet formation did not stop and comparing the pictures before and after the dispensing test shows similar droplet formation.

The results of the measurements of angle deviation of main and satellite droplets, *α_MD_* and α_SD_, (Figure 8) as well as the droplet volume *V_D_* (Figure 9) give more details about the affection of droplet formation. The bar charts show the means ± standard deviations for the state before and after each dispensing test for all four time periods t studied. The values are averaged for all ten experimental replicates. As a result, it can be seen that no significant deviation between *α_MD_* and *α_SD_* occurs in any of the ten experimental replicates for EtOH10ASA at the time periods of t = 1 min and t = 4 min, although drug deposits can be found on the printhead tip, especially at t = 4 min (see Figure 6B,C). For the time period of t = 15 min, the mean value of *α_MD_* does not change, but a negative affection of reproducibility of *α_MD_* is indicated by an increase of the standard deviation. If the time period is extended further to t = 30 min, the droplet formation disturbance increases significantly. Even though droplet formation still occurs in 2 out of 10 experimental replicates, the mean value of *α_MD_* increases sharply and shows a high standard deviation. If satellite droplets occur, no significant influence on the parameter *α_SD_* can be detected for all time periods investigated. The satellite droplets always remain in line with the main droplets. For DMSO100ASA, no negative effects on *α_MD_* and *α_SD_* were observed even at t = 30 min. This is plausible since there were no drug deposits on the tip of the printhead.

It should be noted that the specific deviation angles occur for both drug solutions. Nevertheless, high values of deviation angles are not necessarily critical. If they are stable and reproducible, they can be compensated for by adjusting the printhead alignment, which is a common measure for inkjet printers [49].

As shown in Figure 9 for EtOH10ASA, up to a time period of t = 15 min, reproducibility of *V_D_* is not impaired, which can be seen from the fact that the mean values and standard deviations do not change significantly. Even though droplets still form in 2 out of 10 experimental replicates at the time period of t = 30 min, the mean value of *V_D_* decreases as the nozzle becomes more and more clogged due to the formation of the drug deposits. As a consequence, the diameter of the nozzle of the printhead, and thus the V_D_, decreases. Analogous to the results for α_MD_, no negative influence of *V_D_* was found for DMSO100ASA even at t = 30 min. That is plausible due to the fact that no drug deposits were found on the tip of the printhead. These results show that *V_D_* is basically not affected as much and as quickly as α_MD_. This suggests that initially there is a loss of accuracy in droplet positioning, while the precision of volume dosing is affected less severely and less rapidly. These results are particularly important when high droplet positioning precision is required, as it is the case for drug loading in reservoir-based drug delivery systems (e.g., drug-eluting reservoir stents [50,51] or reservoir-based drug delivery micro-devices for oromucosal drug delivery [21,22] or microneedle systems for transdermal drug delivery [16,17,18].

These results indicate that the nozzle of the printhead does not necessarily clog when drug deposits are present on the tip of the printhead. Despite the intense drug deposition in the case of the time period t = 15 min (see Figure 6D), *V_D_* does not decrease. This indicates that no significant reduction of the nozzle diameter takes place. One reason for this may be that the drug deposits usually form widely around the nozzle because of the creeping effect. According to Townsend et al.’s definition of creeping [37], drug crystal formation begins on the nozzle edge, as this is the line of contact between the drug solution and the nozzle tip of the printhead. As the crystallization process progresses, primary creeping initially dominates. The crystalline structures spread far over the surface of the printhead tip. A significant reduction of the nozzle diameter does not occur since most of the crystalline structures spread towards the outer area around the nozzle. When creeping occurs (especially primary creeping), it is typical that most crystalline structures grow from the boundary of the liquid phase and spread away from the center of the liquid phase during the growth process [37,39]. For this reason, the drug deposits are widespread on the printhead tip without clogging the nozzle. However, due to the continuous growth of crystalline structures, the wetting conditions at the nozzle are constantly changing. For this reason, the conditions for the droplet pinch-off change frequently, which negatively affects the reproducibility of the droplet trajectory, as found for α_MD_. With an increasing time period up to t = 30 min, the amount of drug deposits grew and after a certain amount of time, the nozzle of the printhead clogged. The reason is the occurrence of secondary creeping. Due to the large area of the high amount of drug deposits, which has to be wetted by new drug solution, secondary creeping becomes dominant and further crystals grow onto previously deposited crystalline structures after a certain period of time. This leads to crystalline structures growing not only over the area around the nozzle, mostly directed away from the nozzle center (primary creeping), but more intensive in all spatial directions (secondary creeping), which is in agreement with the findings of van Enckevort and Los [38]. As a consequence, complete clogging of the nozzle occurs after a certain amount of time.

Comparing the results of the experiments with EtOH10ASA with the results of the experiments with EtOH100ASA, it is noticeable that droplet formation impairment occurs later with EtOH10ASA (after 15 min) than with EtOH100ASA (after a few seconds). This is quite plausible in view of the different drug concentrations (EtOH10ASA: c = 10 g/L, EtOH100ASA: c = 100 g/L). The drug deposits we found for EtOH10ASA appear to be much larger than for EtOH100ASA (compare Figure 6D and Figure 2A4). However, the negative effects on droplet formation are comparable (no clogging of the nozzle but a loss of droplet trajectory reproducibility). This seems to be contradictory, but in the case of EtOH10ASA, the drug deposits grow widespread over the area of the tip of the printhead around the nozzle without clogging the nozzle itself for a relatively long time. In the case of EtOH100ASA, the growth of drug deposits is much faster due to the higher drug concentration in the solution. The growth appears to be so rapid that the crystalline structures do not spread as far across the tip of the printhead as is the case with EtOH10ASA. As Enckevort and Los describe with respect to creeping, a supply of new solution is needed that the growth process of the crystalline structures continues. The new solution is supplied by a flow of liquid along the crystals directed to the boundary of the crystalline structures [38]. Due to the lower drug concentration in EtOH10ASA, the formation of ASA crystals occurs less rapidly than for EtOH100ASA due to solvent evaporation. As a result, the time of liquid flow along the crystals is longer for EtOH10ASA than for EtOH100ASA, resulting in a wider spreading of the drug deposits over the tip of the printhead. Primary creeping outweighs secondary creeping for a relatively long time. Most drug deposits form widespread around the nozzle but usually do not spread into or over the nozzle. As a result, relative widespread and conspicuous drug deposits form before the nozzle clogs. In the case of EtOH100ASA, drug deposit formation due to solvent evaporation occurs much more rapidly because of the higher drug concentration in solution. As a result, liquid flow over the deposited crystalline structures is limited because new crystals form too quickly on already deposited crystalline structures. Consequently, there is a lack of new drug solution at the boundary of the deposited crystalline structures and the spread of crystalline structures over the surface of the tip (primary creeping) becomes less dominant. More crystals form and grow onto previously deposited crystalline structures (secondary creeping). Secondary creeping quickly becomes dominant over primary creeping, and a relatively large ASA crust forms directly on the tip of the printhead within a few seconds (Figure 2), which is not as widespread as for EtOH10ASA (Figure 6). In contrast to EtOH, the solvent DMSO allow a high drug concentration of c = 100 g/L in our study without negative effects on droplet formation, even at a relatively long time period of t = 30 min. Due to the low volatility of DMSO, the drug solution DMSO100ASA did not cause drug depositions on the printhead tip. Consequently, EtOH10ASA enables long-term inkjet printing without the risk of losing droplet reproducibility after a few minutes. However, inkjet printing must be frequently interrupted to perform a washing procedure to clean the printhead tip. DMSO100ASA does not require an interruption of inkjet printing to perform a washing procedure, even at relatively high inkjet printing durations, as shown in our study. These results suggest that low volatile solvents such as DMSO are preferable to highly volatile solvents such as EtOH for long-term inkjet printing of highly concentrated drug solutions. However, our study is limited to droplet formation. The volatility of solvents must also be considered when drying the droplets after printing on a substrate to obtain the API in the desired solid form [26,29]. Since solvents can have toxic side effects, residue-free evaporating of the solvents is required. This can be a limiting factor especially in the selection of low volatile solvents. To adjust the evaporation properties of a drug solution, it is also possible to mix various solvents of different volatilities. For example, Scoutaris et al. used a mixture of EtOH and DMSO (95/5) to prepare a drug solution (containing the API Felodipine) with an elevated boiling point [7]. The authors did this to reduce the risk of clogging the dispenser used for inkjet printing investigations. However, when using solvent mixtures, the solubility of the desired API in the solvent mixture must be considered. In addition, the use of highly volatile solvents in inkjet printing can be beneficial to minimize fabrication time [35].

## 4. Conclusions

In this study, we investigated inkjet printability and the long-term reproducibility of droplet formation of ASA drug solutions based on the solvents EtOH and DMSO. Typical parameters of droplet formation (droplet volume, droplet speed, and angle deviation of the droplet’s trajectory) were investigated. The dimensionless numbers *Z* (inverse of Ohnesorge number *Oh*) and Weber number *We* were used to evaluate the experimental results of inkjet printing.

There are significant differences in droplet formation between EtOH- and DMSO-based drug solutions. Due to the different fluid parameters, especially viscosity and surface tension, DMSO-based drug solutions require a higher printhead voltage to generate droplets with a similar droplet volume. We found that dissolving relatively large amounts of ASA in the solvents EtOH and DMSO (c = 100 g/L) had a relatively small effect on the fluid parameters of the drug solutions. As a result, the droplet parameters droplet volume and droplet speed are quite similar when using the same voltage for the drug solutions and for the pure solvents. These findings are consistent with the calculated *Z* and *We* numbers. These numbers are also quite similar for drug solutions and pure solvents.

However, the different volatilities of the drug solutions lead to differences in droplet formation. Due to their high volatility, the EtOH-based drug solutions EtOH100ASA (relatively high concentrated with c = 100 g/L) and EtOH10ASA (relatively low concentrated with c = 10 g/L) showed an increase in drug deposition of ASA on the nozzle, which affected droplet formation. The droplet formation parameters droplet volume V_D_, droplet speed *u_D_* and droplet trajectory (deviation angles α_MD_, α_SD_) are influenced differently. The reproducibility of droplet trajectory, characterized by the deviation angles *α_MD_* and α_SD_, is negatively affected the most. This is a major factor limiting the accuracy of droplet deposition. The reproducibility of droplet volume *V_D_* and droplet speed *u_D_* is not affected as much. Steady growth of drug deposits leads to complete clogging of the nozzle after a certain amount of time. The higher the concentration of the EtOH-based drug solution, the faster the drug deposits grow and the faster negative effects occur in droplet formation. The droplet formation of EtOH100ASA was negatively affected after only a few seconds, while the droplet formation of EtOH10ASA was largely stable up to a time period of t = 15 min. We have found that the grade of negative influence on droplet formation does not necessarily correlate with the amount of drug deposits on the tip of the printhead. Due to the creeping effect during crystallization, the crystalline structures can grow on the surface around the nozzle without clogging the nozzle for a certain amount of time. In contrast, the very low-volatile but highly concentrated (c = 100 g/L) drug solution DMSO100ASA did not show any growth of drug deposits even after the long time period of t = 30 min.

We conclude that drug deposits on the printhead nozzle do not necessarily have a negative influence on droplet formation. If there is a negative influence, there is initially a loss of droplet positioning accuracy. The precision of volume dosing is not affected as much or as quickly. However, the growth of drug deposits leads to clogging of the nozzle after a certain amount of time. In addition, for long-term inkjet printing of highly concentrated drug solutions, low volatile solvents such as DMSO are generally preferable to highly volatile solvents such as EtOH. When using highly volatile solvents, inkjet printing must be interrupted frequently to perform a washing procedure to clean the printhead tip. When using low volatile solvents, the inkjet printing may not need to be interrupted to perform a washing procedure, even with relatively long time periods and relatively high drug concentrations. These results are of particular relevance when high droplet positioning precision is required, as it is the case, for example, for drug-loading in reservoir-based drug delivery systems as well as drug-loading of microneedle systems.

Our study is limited to the droplet formation. Nevertheless, the solvent volatility is also a key factor in the drying of the droplets after printing on a substrate. Consideration of droplet evaporation rate after printing is vital to obtain the API in the desired solid form. Further investigations need to focus on droplet drying and drug formulation on a substrate.

## Figures and Tables

**Figure 1 pharmaceutics-15-00367-f001:**
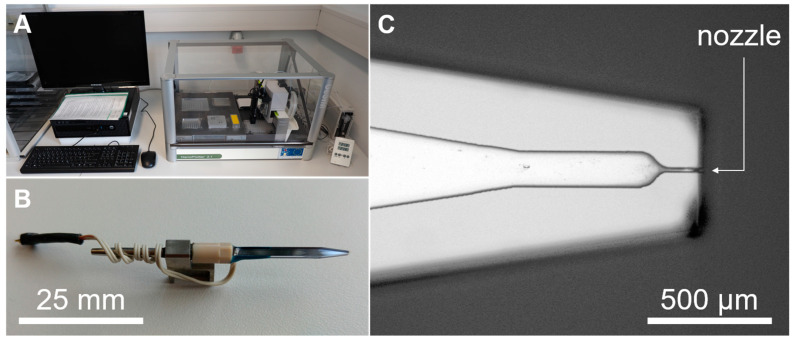
(**A**) Nanoplotter 2.1 device (GeSiM mbH, Radeberg, Germany). (**B**) Piezoelectric DOD inkjet printhead model “Pico-Tip J” for Nanoplotter 2.1 device (GeSiM mbH, Radeberg, Germany). (**C**) Microscopic picture of the tip of the printhead with the nozzle (microscopic imaging system: BX51, Olympus K.K., Tokyo, Japan).

**Figure 2 pharmaceutics-15-00367-f002:**
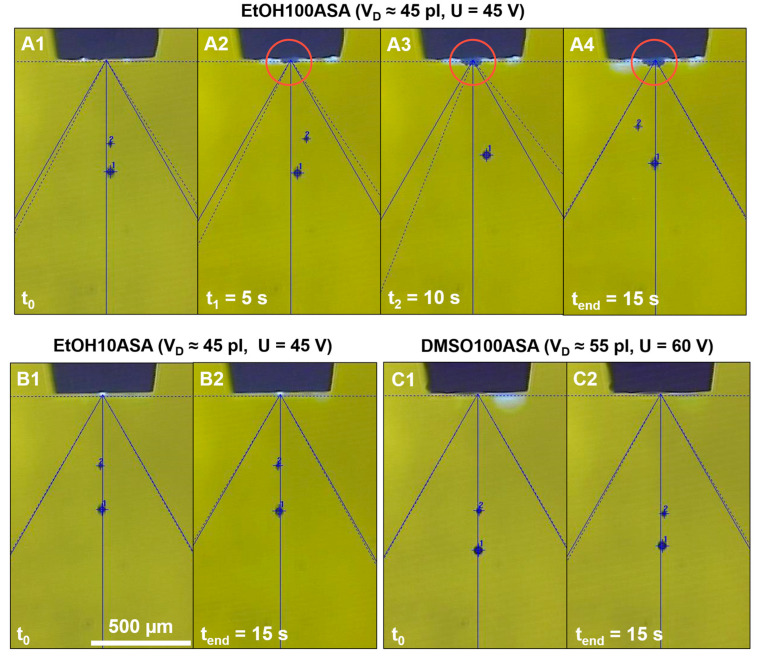
Exemplary stroboscopic image capturing system photographs of droplet formation of the drug solutions EtOH100ASA (**A1**–**A4**), EtOH10ASA (**B1**,**B2**) and DMSO100ASA (**C1**,**C2**) over time of the investigation. Droplet formation of EtOH100ASA is interfered, as there is a steady growth of drug deposits on the nozzle (circled red). For EtOH10ASA and DMSO100ASA there are no notable drug deposits and a highly reproducible droplet formation.

**Figure 3 pharmaceutics-15-00367-f003:**
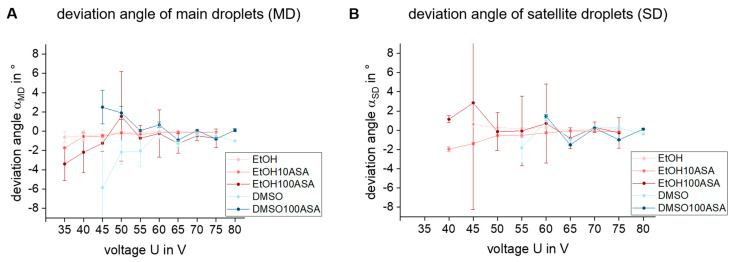
Plots of deviation angle of main droplets (α_MD_, (**A**)) and satellite droplets (α_SD_, (**B**)) over voltage of the printhead for EtOH-based drug solutions as well as the pure solvents. The deviation angle of a main droplet (α_MD_) was measured with regard to a vertical line from the nozzle, the deviation angle of a satellite droplet *α_SD_* was measured with regard to the trajectory of the associated main droplet. There are relatively high standard deviations for EtOH100ASA over the whole voltage range. Therefore, there is low reproducibility of a droplet’s trajectory. The plots show means ± standard deviations for *n* = 10.

**Figure 4 pharmaceutics-15-00367-f004:**
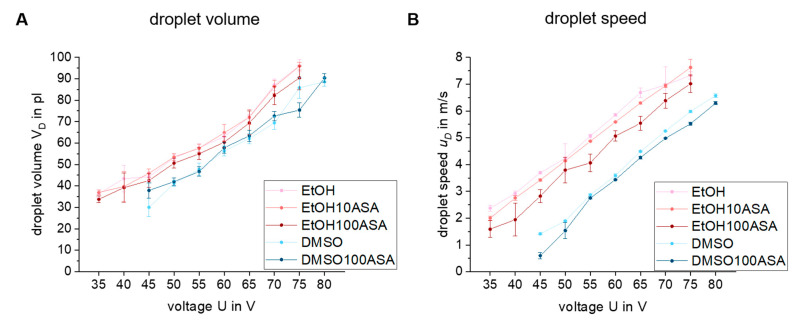
Plots of the droplet volume *V_D_* (**A**) and droplet speed *u_D_* (**B**) over voltage of the printhead of all investigated drug solutions as well as the pure solvents. Drug solutions and pure solvents show similar averages and standard deviations. There is a little decrease of *V_D_* and *u_D_* of EtOH100ASA as a result of the increase of viscosity by drug solving. The plots show means ± standard deviations for *n* = 10.

**Figure 5 pharmaceutics-15-00367-f005:**
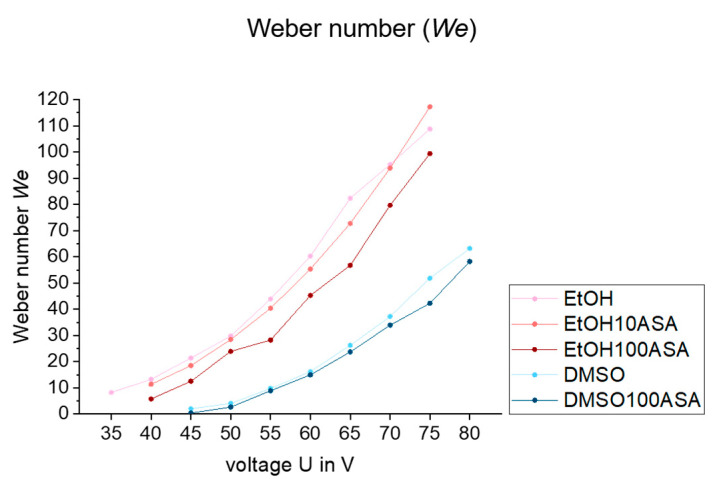
Plot of Weber number *We* over voltage for all investigated drug solutions as well as the pure solvents. A *We* < 40 is required to avoid splashing when droplets hit (wetted) substrates [32]. This *We* value can be achieved for all drug solutions if a suitable voltage is selected.

**Figure 6 pharmaceutics-15-00367-f006:**
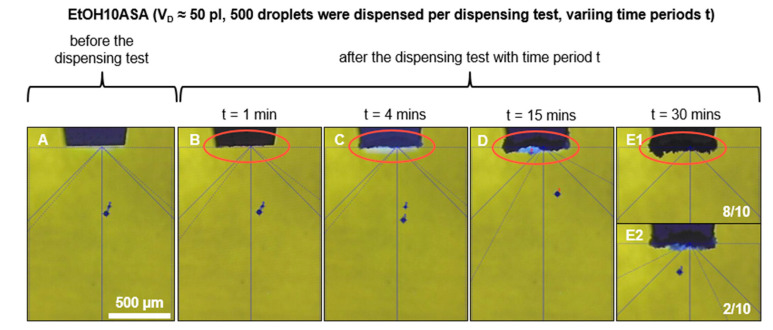
Droplet formation before (**A**) and the end of the dispensing tests with different time periods (**B**–**E**). 500 droplets of the drug solution EtOH10ASA were ejected uniformly over each of the four different time periods. Drug deposition increases with increasing time period (circled red). Clogging of the nozzle and cessation of droplet formation were detected only for the time period t = 30 min at eight out of ten experimental replicates (**E1**). Droplet formation still occurred in 2 out of 10 experimental replicates but was significantly impaired (**E2**).

**Figure 7 pharmaceutics-15-00367-f007:**
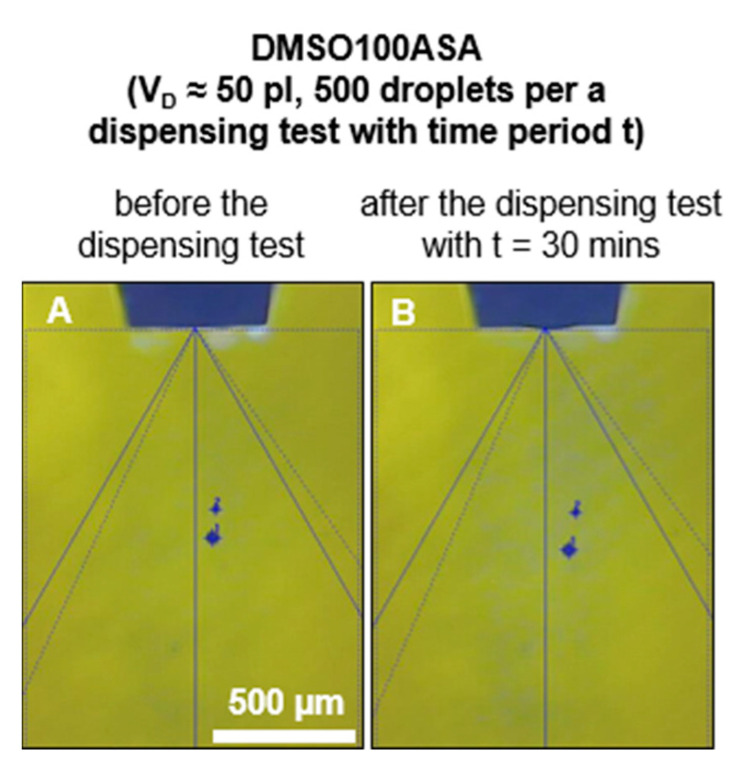
Droplet formation at the beginning (**A**) and end of the dispensing test with time period t = 30 min (**B**). 500 droplets of the drug solution DMSO100ASA were ejected uniformly over the time period. Despite the high drug concentration of c = 100 g/L, no drug deposits were detected even for the longest time period of t = 30 min.

**Figure 8 pharmaceutics-15-00367-f008:**
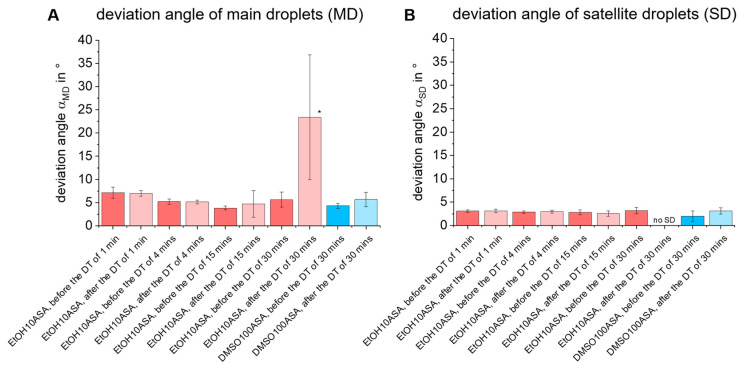
Plots of (**A**) deviation angle of main droplets (α_MD_) and (**B**) satellite droplets (*α_SD_*) of EtOH10ASA (c = 10 g/L ASA) and DMSO100ASA (c = 100 g/L ASA) before and the end of the dispensing tests (DT) with different time periods of t = 1 min, t = 4 min, t = 15 min and t = 30 min. 500 droplets of the drug solution EtOH10ASA were ejected uniformly over each of the four time periods. At a time period of t ≥ 15 min, droplet formation in EtOH10ASA is negatively affected by drug deposits on the printhead tip whereas the mean values of *α_MD_* do not change significantly. However, the reproducibility of *α_MD_* decreases, as indicated by an increase in the standard deviation. At a time period of t = 30 min, there is a significant increase in the mean value and standard deviation of *α_MD_* and no satellite droplets (“no SD”) were found (* the measurements were carried out at 2 out of 10 experimental replicates, termination of droplet formation occurred in 8 out of 10 experimental replicates). The deviation angle of the satellite droplets *α_SD_* was not negatively affected, but at a time period of t = 30 min, satellite droplet formation stopped during the dispensing test. For DMSO100ASA mean values and standard deviations of *α_MD_* and *α_SD_* did not change significantly. Droplet formation was not negatively affected because there were no drug deposits on the printhead tip even at the time period of t = 30 min despite the high drug concentration of c = 100 g/L. The plots show means ± standard deviations for *n* = 50.

**Figure 9 pharmaceutics-15-00367-f009:**
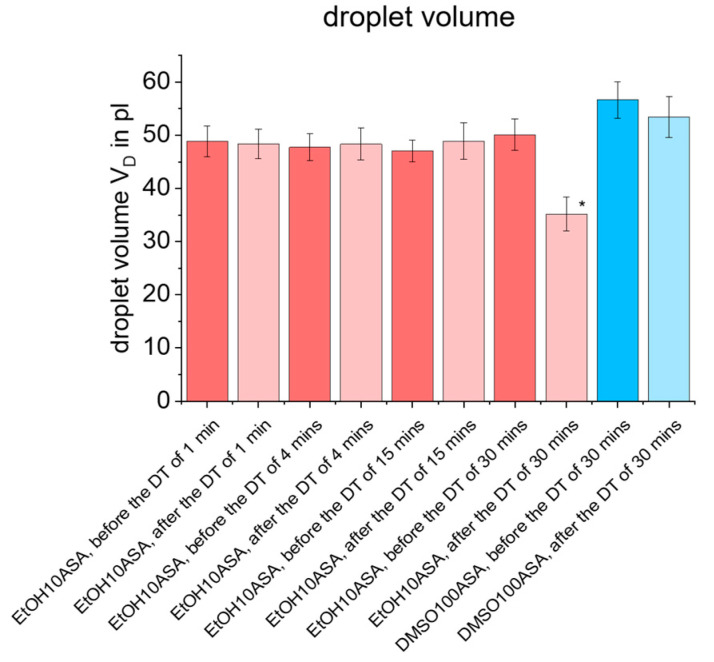
Plot of droplet volume *V_D_* for EtOH10ASA (c = 10 g/L ASA) and DMSO100ASA (c = 100 g/L ASA) before and after the dispensing tests (DT) with time periods t = 1 min, t = 4 min, t = 15 min and t = 30 min. In each time period, a number of 500 droplets was ejected uniformly. At a time period of t = 30 min, EtOH10ASA droplet formation is negatively affected by drug deposits on the printhead tip. The mean value decreases due to nozzle clogging caused by drug deposits (* the measurements were carried out at 2 out of 10 experimental replicates, termination of droplet formation occurred in 8 out of 10 experimental replicates). For DMSO100ASA, mean values and standard deviations did not change significantly. Droplet formation was not negatively affected, as there were no drug deposits on the printhead tip even at a time period of t = 30 min. The plot shows means ± standard deviations for *n* = 50.

**Table 1 pharmaceutics-15-00367-t001:** Overview and details of the investigated drug solutions.

Name of Drug Solution	Solvent	API	c in g/L
EtOH10ASA	EtOH	ASA	10
EtOH100ASA	EtOH	ASA	100
DMSO100ASA	DMSO	ASA	100

**Table 2 pharmaceutics-15-00367-t002:** Overview of averaged values of measured fluid properties (means ± standard deviations, *n* = 3 each), the nozzle diameter and calculated *Z* numbers. *Z* numbers are suitable for inkjet printing. The appearance of satellite droplets is likely.

Medium	*ρ* in g/cm^3^	*σ* in mN/m	*η* in mPas	*Z*
DMSO	1.09930 ± 0.00001	41.52 ± 0.27	2.06 ± 0.06	17.67
DMSO100ASS	1.11314 ± 0.00002	42.14 ± 0.44	2.42 ± 0.06	15.24
Ethanol	0.78866 ± 0.00001	22.12 ± 0.15	1.17 ± 0.06	19.23
ETOH100ASS	0.82977 ± 0.00001	22.89 ± 0.14	1.28 ± 0.05	18.35
ETOH10ASS	0.79288 ± 0.00001	22.32 ± 0.10	1.13 ± 0.01	20.04

## Data Availability

The raw data required to reproduce the findings of this article are available from the corresponding author upon reasonable request.

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
