# Peer review of "Influence of the Volatility of Solvent on the Reproducibility of Droplet Formation in Pharmaceutical Inkjet Printing"

_pharmaceutics, 2023, doi:10.3390/pharmaceutics15020367_

Round 1

Reviewer 1 Report

In the manuscript, the authors studied the inkjet printability and long-term reproducibility of droplet formulation. For tests, solutions of acetylsalicylic acid were prepared using ethanol (two different concentrations) and DMSO as solvents. The authors investigated the parameters of droplet formation, such as droplet volume, droplet speed, angular deviation, and droplet trajectory. The authors used the dimensionless Z and We formula to evaluate the experimental results.

The manuscript is well-edited. 

The authors used significant and relevant references. The figures and tables are informative.

A few small additions:

1. Standard deviation values are shown in Table 1 but in 2.2. chapter, the numbers of the parallel measurements are not given in the description of the measurements.

2. It is not specified which microscope the authors used in the case of Fig1/C.

3. In my opinion, in the case of Fig. 3; 4 and the bar charts, the number of parallels should also be specified, because they show SD values

Reviewer 2 Report

In this manuscript, the long-term reproducibility of droplet formation of piezoelectric inkjet printed drug solutions using solvents with different volatilities was investigated. The results showed that the repeatability of droplet formation with the low volatile solvent was preferable to the high volatile solvent. The issues discussed in this work are meaningful, however, the manuscript is not well written and the novelty of this research is unclear. This paper is not recommended for publication in this journal.

The authors need to pay attention to the following issues:

1.The manuscript is not carefully treated.

Firstly, the introduction is not logical enough. Secondly, this manuscript lacks the clear novelty and/or significance required for publication.

2.Further work plans are not necessary in the abstract.

3.Page 5, line 190,  the droplet volume vd represents the volume sum of main and satellite droplets. Why is it calculated in this way?

4.Page 5, what is it below in Figure 1(B)?  It can be removed if not necessary.

5.Page 8, the drug deposits minimize the nozzle diameter by comparing the changes of viscosity, droplet volume, and droplet velocity between pure solvent and solution. Thats not rigorous enough. It is suggested to compare the droplet volume and velocity of the two solutions at different times.

6.Page 13, due to the creeping effect, the drug is deposited in the area around the nozzle at t=15min and the nozzle is clogged by secondary creeping at t=30min. Additional evidence is needed to make the interpretation more convincing.

7.Whether EtOH solvent or DMSO solvent, there are satellite droplets in the printing process, which greatly affects the precision of inkjet printing, and its application value will be greatly reduced. Therefore, it is recommended to adjust printing parameters to make the injection more stable.

8.Some improprieties exist in the manuscript, for example but not limited to this: 

the unit of surface tension is mN/m not nNm in Table 2, the serial numbers are lost in Figure 3 and Figure 4, the information of ref.14 and ref. 18 is incomplete.

9.The conclusion is not a simple summary of results and discussion, it is refined and generalized. Please polish the conclusion further.

Reviewer 3 Report

The review paper summarizes the DOD inkjet printability and long-term reproducibility of droplet formation of API in various solvents.  

It is a convincing and valuable fundamental paper for using inkjet printing technology in the pharmaceutical area. 

Only some minor comments for this paper:

Section 2.4 Investigation of inkjet printing parameters: The humidity of the environment is an essential parameter for inkjet printing. It will be better to add the information at the end of this section after the temperature.

Statistical analyses are required to compare all the quantitative data. This is important since lots of process parameters were changed and compared. Please summarize the details in one section in the materials and methods part.

Results:

How about the prospect of using various solvents with various APIs by using multiple solvents? 

Round 2

Reviewer 2 Report

The authors have well responded to reviewers’ comments. The manuscript has been greatly revised and the English has been improved. The introduction section has been revised for further clarifying the novelty of this manuscript. Meanwhile, the layout of this manuscript is proper, the figures are acceptable, and the section of the results and discussions is substantial. Overall, the revised manuscript can be basically accepted for publication in this journal.